# Level of Estrogen in Females—The Different Impacts at Different Life Stages

**DOI:** 10.3390/jpm12121995

**Published:** 2022-12-02

**Authors:** Zhuo Yu, Yan Jiao, Yinhuan Zhao, Weikuan Gu

**Affiliations:** 1Heilongjiang Academy of Traditional Chinese Medicine, Sanfu Road 142, Xiangfang District, Harbin 150040, China; 2Department of Orthopedic Surgery and BME-Campbell Clinic, University of Tennessee Health Science Center, Memphis, TN 38163, USA; 3Department of Rheumatism, Shanghai Traditional Chinese Medicine Integrated Hospital, Shanghai University of Traditional Chinese Medicine, 230 Baoding Road, Hongkou District, Shanghai 200082, China; 4Research Service, Memphis VA Medical Center, 1030 Jefferson Avenue, Memphis, TN 38104, USA

**Keywords:** aging, disease, estrogen, health, lifespan, ovariectomy

## Abstract

Historically, a high level of estrogen in women is regarded as the signature for a longer lifespan than men. Estrogen is known to be responsible for the development and regulation of the female reproductive system and secondary sex characteristics. Ovariectomy brings on numerous complications such as early menopause, heart disease, and osteoporosis. Thus, ovariectomy impacts the long-term health and lifespan of women. However, the level of estrogen at different life stages should be managed differently. Life quality can be measured in many ways, but mainly it relates to how an individual is doing in terms of being healthy, comfortable, and able to participate in or enjoy life experiences. First of all, ovariectomy not only reduces the level of estrogen but also destroys the reproductive metabolism and potentially other metabolism functions; it may also reduce the lifespan because of the overall impact, not necessary due to the low level of estrogen. Secondly, according to the principal law of the lifespan (PLOSP), the impacts of ovariectomy at different life stages will be different. The objective of this article is to provide readers with a new view of the research on estrogen. Based on the PLOSP, we recapture the estrogen levels at different life stages and explore potential alternative approaches to the manipulation of the levels of estrogen based on the biological features of the difference life stages. Thus, a low level of estrogen in the early life stage may make a woman live longer than a woman with a normal level of estrogen. However, a low estrogen level does not equal ovariectomy. Here, we explain the different impacts of the estrogen levels during different life stages; the effects on the lifespan of the manipulation of estrogen levels at different life stages; and the differences among the estrogen levels, ovariectomy effects, life stages, and lifespan. The personalized manipulation of estrogen levels and relevant growth factors according to the characterization of the life stages may be able to extend the heathy lifespan of women.

## 1. Introduction

The sex features of humans are mainly regulated by sex hormones. For women, estrogen plays an important role in their normal sexual and reproductive development. Estrogen also has non-reproductive functions in extragonadal tissues including the liver, heart, muscles, bones, and brain. Thus, a balanced level of estrogen throughout life is essential for the health of women [1]. While the secretion of testosterone or other male-specific hormones has been suspected to be the factor that leads to the shorter lifespan in men [2], estrogen in women has been considered the factor responsible for the longer lifespan in women [3]. 

A woman’s ovaries make most of their estrogen. The importance of estrogen has been shown by the effects of ovariectomy in women [4]. In the human population, the data on ovariectomy are mostly from women with breast cancer [5]. The complications caused by ovariectomy in animals are well-recognized [6]. The focus in most current research studies is on the treatment of the complications of patients who are ovariectomized [7]. Thus, unlike studies on the effects of orchiectomy, which may lead to a longer lifespan, the effects of ovariectomy have been regarded as negative factors for the human lifespan. The important question is whether ovariectomy at any life stage causes more damage than benefit. 

A new theory called the principal law of the lifespan (PLOSP) states that every life stage has its own specific physiological and metabolic characteristics. Each life stage can be lengthened by either slowing its processes or continuously maintaining the activities of its function [8]. Based on the PLOSP, different strategies should be applied to the different life stages and the same strategy may lead to opposite effects when applied to the different life stages [8]. The effects in cancer patients may not reflect the effects in all life stages from ovariectomy, as the majority of these patients are adults. The effects of ovariectomy before puberty are literally unknown. Here, we discuss the potential bias in the past studies of estrogen and explain the possible extension of the lifespan of females by slowing the development of estrogen in the early life stages, as well as the benefit of managing the levels of estrogen based on the life stages.

## 2. Sex Differences in Lifespan and Estrogen

### 2.1. The Observed Sex Differences in the Lifespan 

It is a well-recognized fact that males have a shorter lifespan than females [9,10,11] (Figure 1). In animals, the lifespan of males in most cases is much shorter than that of females. In all countries, the published life expectancy rates for women are all longer than for men. Because this phenomenon comes from a sex difference, decades ago, when the research was in the early stage and the resources and data were limited, the reason for this difference was automatically tracked to the features of sex hormones, while the molecular mechanism is still not completely understood. Although this difference may be influenced by many other factors, such as gender-associated risk factors and physical and sociological differences, the level of testosterone has been regarded by a considerable number of researchers as the cause that leads males to live shorter lives than females [2,9,10]. One risk factor related to testosterone in the lifespan of males is that it is connected to their aggressiveness [9,10]. Following this idea, estrogen, on the other hand, has been considered as the factor that leads to the longer lifespan in females. The functions of estrogen in the brain are mainly to do with anxiety and depression, not aggression [11]. These apparently correct concepts have widely influenced the biomedical research tremendously, with publications assessing the levels of testosterone and estrogen via orchiectomy and ovariectomy. One factor that has been ignored in these studies is the life stages [8,12]. 

### 2.2. The Paradigm of the Impact of Low Estrogen Levels on the Healthy Life of Females

Except for the abnormal high and low levels of estrogen, healthy women in general have much higher levels of estrogen than the men. Unlike the performance of orchiectomy on men, historically ovariectomy has not been applied to healthy women. The impact of the maintenance of a high level of estrogen is recognized mainly from patients who are ovariectomized for the purpose of disease treatment. The complication of the ovariectomy of adult women is described as involving rapid aging [13], including early menopause, depression, heart diseases, memory problems and osteoporosis [5,6,7,13]. As such, the low level of estrogen resulting from ovariectomy in women has been regarded as the factor for complications and uncoupling of the lifespan [13].

In connection with the observation that women live longer than men, the complications caused by ovariectomy strengthen the idea that estrogen is the key hormone that makes women live long. However, almost all of the data on ovariectomy are based on women during adulthood, not girls before puberty. There is a knowledge gap on whether the same complications will be brought about by ovariectomy in females before puberty and at different life stages. Age differences have been shown but have not been recognized as the difference in the life stages [13].

### 2.3. Confusing Data from Animal Models on the Function of Estrogen

Obviously, animal models are particularly suitable for testing the effects of estrogen on the lifespan, as ovariectomy can be performed to reduce the production of estrogen (Table 1). Thus, ovariectomy has been used in animal models as a tool to test the effects of the group of hormones classed as estrogen. Similar to the study on orchiectomy, the study on ovariectomy did prove that a reduction in estrogen may shorten the lifespan [14,15]. Several typical reports caused confusion regarding this aspect. Apelo et al. [14] reported that ovariectomized females showed improved survival despite paradoxically showing increased adiposity and decreased control of blood glucose levels. A notable feature in this study was that the mice were gonadectomized or subject to sham surgery during the third week of life, most likely before puberty. Tetlak et al., reported that ovariectomy led to an increased lifespan, increased protein storage, and decreased feeding in grasshoppers [15]. In spite of the facts that emerged from the studies on grasshoppers and rats, other factors may have affected the effects of ovariectomy [15], and together these studies raise doubts regarding the effects of estrogen reductions on the lifespan. An interesting observation on the desexing of dogs suggested that ovariectomy before puberty may benefit the lifespan of female dogs. In a review of the documents on the desexing of dogs, Urfer and Kaeberlein concluded that the beneficial effect of desexing is stronger in female than in male dogs [16]. It is suspected to be because the time of puberty in female is later than that in male dogs.

In contrast, an early study found that ovariectomy shortens the life span of female mice [19]. When we looked at the detailed experiment procedure in this study, we found that two groups of mice were ovariectomized. In one group, the mice were subjected to ovariectomy at 1 month, after the first estrous cycle, while in the other group, the mice were subjected to ovariectomy at 5 months of age. Obviously, the ages of the mice in this study were after puberty. This was likely the reason that led to the different results as compared to the study by Apelo et al. [14]. 

Several studies using aged mice have shown that the age of the mice is important. Three studies using aged mice showed that ovariectomy reduces the lifespan of the mice [19,20,27]. The other early study using mice at 4 weeks showed that ovariectomy increases the squamous metaplasia of the uterine horns and survival [22]. 

However, the effects of ovariectomy at the different life stages are subject to mouse strain differences, especially genetic variation. One study using young animals of the CBA strain also indicated that ovariectomy decreases the lifespan of mice [23]. However, we noticed that the CBA/J inbred mouse strain was used to study granulomatous experimental autoimmune thyroiditis (G-EAT), and CBA/J mice are relatively resistant to diet-induced atherosclerosis [24].

In comparison to the mouse models, in the studies using rats, almost all studies applied the ovariectomy to aged rats [25,26]. Majority of the studies are also focused on the treatment of the complications of ovariectomy. 

## 3. Regulation of the Level of Estrogen to Increase the Lifespan Based on the PLOSP

### 3.1. Explanation of Effects of Estrogen at Different Life Stages Based on the PLOSP

Based on the PLOSP [8], the human lifespan can be extended to different life stages, including the body’s growth stage before puberty, the reproductive stage between puberty and menopause [12], and the aging stage after menopause. Accordingly, our hypothesis is that the manipulation of the levels of estrogen in the different life stages may impose different effects on the lifespans of the women. 

Thus, when using ovariectomy as a technique to manipulate the levels of estrogen in the study of animal models, one must consider the life stages (Figure 2). Studies have shown that the ages of puberty of humans have decreased among all ethnic and geographic human populations, partially because of changes in environment and nutrition [12]. During the body’s growth stage, the level of estrogen changes from none at all and begins to appear and increase. A rapid increase in the level of estrogen leads to early puberty. At this stage, slowing or reducing the level of estrogen may not impose negative impacts on the human life and may extend the lifespan [12]. It is expected that animal studies will show that slowing down the speed of hormone level increases during the body’s growth stage will extend the length of the period between birth and puberty. In other word, it will increase the length of the body’s growth stage. The data then can be utilized in the human population to manipulate the level of estrogen and other factors to extend the body’s growth stage.

During the reproductive stage between puberty and menopause, disrupting and dramatically reducing the estrogen production will destroy the normal metabolism of the reproductive system. The ovariectomy in this case may push the life cycle to enter the aging stage [12]. 

At the aging stage, the level of the estrogen has decreased, and further decreasing its production may speed up the aging, although the effect may not be as great as that of the ovariectomy during the reproductive sage [8,12]. 

### 3.2. Differences between Ovariectomy and Orchiectomy in Different Life Stages

When considering the levels of estrogen at the different life stages, one also needs to understand the techniques involved in the manipulation of the level of estrogen and the similarities and differences to orchiectomy. The current information raises the question on whether the impacts of orchiectomy and ovariectomy on the lifespan are the same. Although these two techniques are used to reduce the levels of sex hormones, there are two differences between the effects of orchiectomy and ovariectomy. In the human population, ovariectomy causes more complications than orchiectomy. In animal models, orchiectomy at young ages extends the lifespan, while ovariectomy in a previous study showed either similar results with non-significant differences or even shortened the lifespan [23]. Furthermore, the differential effects on young, middle-aged, and aged female mice via life-long environmental enrichment support the much wider impacts of ovariectomy on the metabolism and lifespan of female mice [22]. The other example is that Bame et al., reported that methionine sulfoximine treatment improves the survival rates of both male and female mice, but ovariectomization or orchiectomy eliminates the effect of the drug on either survival or neuromuscular deterioration [28]. In addition, in men, the levels of testosterone decrease with age while the levels of estrogen increase. This information challenges the original theory that male hormones make men live shorter lives than women; on the other hand, it also raises the question as to whether estrogen in women causes their longer lifespan.

### 3.3. The Impact of Ovariectomy Is Much More than a Decrease in Estrogen 

As shown above, the studies of ovariectomy in animal models have produced controversial results. Its effect on the lifespan is not clear. In the human population, ovariectomy is known to cause multiple side effects such as menopause signs and symptoms, depression or anxiety, and osteoporosis. Overall, the effects of ovariectomy seem complicated. However, the effects of ovariectomy are much more than a decrease in estrogen production. Two additional issues are important to understand. The first one is to distinguish between estrogen and ovariectomy, testosterone and orchiectomy, and orchiectomy and ovariectomy. Ovariectomy involves removing the major organ responsible for the production of estrogen, as well as other hormones and other functions of the ovaries. Similarly, orchiectomy involves removing the major organ responsible for the production of testosterone, as well as other functions of the testicles. Thus, ovariectomy does not equal the depletion of estrogen production, while orchiectomy does not equal a decrease in the production of testosterone. Most importantly, the effects of the ovaries may be greater than those of the testicle on the development of the human body. The locations and structures of the testicles and ovaries are different and the impacts on the human body may also be different. If we believe that the impacts on the body’s growth and development of the ovariectomy are different in women than those of the orchiectomy on men, we then can explain why the ovariectomy shortens the lifespan while the orchiectomy may not, because the effects are not only from estrogen or testosterone.

Secondly, we need to understand that effect of ovariectomy on the balance of estrogen and other hormones. The ovariectomy may not only reduce the production of estrogen but also damage the balance of the human hormone system. Because the levels of estrogen and other hormones are different at different life stages, ovariectomy at different life stages may lead to the biased balance of the hormone system in comparison to the normal balance. One example of such a difference is from the report by Cargill et al. [23]. The authors first ovariectomized mice at a young age. Later, when the same young ovaries were implanted into mice ovariectomized at different ages, the authors found that the length of the life extension was different. In particular, when the young ovaries were implanted into aging mice at 11 months of age, the lifespan was extended the most [23]. 

### 3.4. Manipulation of Estrogen Levels at Different Life Stages Based on the PLOSP

After understanding the complexity of the estrogen levels and ovariectomy, and if we agree with the PLOSP that the manipulation of the levels of estrogen at different life stages may lead to different effects, we can then understand how the changes in estrogen levels may differently affect the different life stages (Figure 3). The first life stage to involve estrogen level changes is between birth and puberty. During this stage, the body grows quickly, and the estrogen level changes from none at all to the maximum level. Because of the critical role of the estrogen level in puberty, it is expected that a low or slowly increasing level of estrogen during the body’s growth stage may delay the time of puberty [8,12]. As the age of puberty in the human population has decreased in the past decades [8,12], the age of puberty can be postponed by lowering the level or slowing down the increase in the level of estrogen, which will most likely extend the lifespan [12,22]. However, clear evidence using animal models to show the effects of ovariectomy before puberty shows that a normal animal strain should be selected because of the complicated effects from interactions between the ovariectomy and animal strain background [21,22,23,24]. The other consideration is that because a bilateral ovariectomy involves the removal of both ovaries, the damage to the body may exceed the benefit of lowing the estrogen level. After clear evidence is obtained from animal studies, a proper clinical trial may be conducted among the human population. While ovariectomy can be used for studies with animal models to test the effects of estrogen and other hormones, for humans, other means must be used for such studies. The manipulation of the estrogen level using molecular methods is most likely the suitable approach. It remains to be seen whether any measures that lower the estrogen level while not removing the ovaries will benefit the lifespan. 

The reproductive stage is between puberty and menopause in women. During this period, the estrogen level decreases slowly. If the estrogen remains at a relatively high level, a woman’s reproductive capability may remain unchanged. Thus, the life stage of reproduction in women may be extended by prolonging the high levels of estrogen and perhaps also using other environmental and nutritional factors. Therefore, methods that increase the level of estrogen production and the balance of the hormone system at this stage can be used to extend the lifespan. There is enough evidence from the human population and animals indicating that lowering or disrupting the estrogen production at this stage will cause many complications and shorten the lifespan [4,5,7,13,19,27]. 

After women enter the aging stage, when the production of estrogen is reduced dramatically, increasing the estrogen level may benefit female health or help women remain in the reproductive stage, thereby elongating the lifespan of the women. Animal studies have confirmed such cases where the lifespans of aged animals were extended with the implantation of young ovaries [23]. 

While the difference between insects and mammals should be recognized, these studies point out that estrogen may not be the cause that leads to the sex differences in lifespans (Figure 4) [15,29,30]. As time goes on, we look forward to seeing the impacts of estrogen in the different life stages in multiple animal models and human populations. 

### 3.5. Hormone Replacement Therapy as Evidence Suporting the Regulation of the Level of Estrogen to Increase the Lifespan Based on the PLOSP

The data on hormone replacement therapy (HRT) and the lifespan of women can be used to support the PLOSP in the human population [31,32,33]. First of all, the application of HRT may benefit the lifespan of women who have undergone ovariectomy. Armstrong et al., reported that HRT is associated with an increase in life expectancy of between 0.39 and 0.79 years for mutation carriers undergoing both prophylactic mastectomy and oophorectomy [31]. Secondly, aging is a factor that influences the effects of HRT. The results from an international comparison among European and North American populations showed that HRT could result in benefits with regard to overall mortality, but this advantage decreases in younger-generation cohorts [32]. Thus, there is a potential life stage difference. 

Currently, HRT also causes multiple side effects. While in general every therapeutic application has side effects, it is possible to reduce the side effects of HRT with the use of improved protocols in the future, such as the individualized application of HRT with different protocols. For example, recently it has been confirmed that thyroid hormone replacement therapy was effective in patients with heart failure and low-triiodothyronine syndrome [34]. Despite the side effects, the fact that overall the lifespan was increased using HRT shows that the regulation of the hormone levels at the right time and right life stage may be used as one tool to extend the lifespan.

## 4. Considerations for the Regulation of Estrogen Based on the PLOSP

### 4.1. Correct Interpretation of Estrogen Levels Based on the PLOSP

One must understand that the regulation of the levels of estrogen at different life stages to extend the lifespan is an option for women with healthy or normal developmental and physiological conditions. The manipulation does not apply to populations with abnormally high or low levels of estrogen. Studies have confirmed that no matter the life stage, extremely low or high levels of estrogen will lead to health problems and a shorted lifespan [26,28,29]. 

Currently, there is a lack of studies on the physiological conditions and biomarkers for life stage transitions and the turning points between the life stages [12]. We did not find any study to compare the life stage transitions and turning points between the life stages, such as the physiological alterations during the transition periods between puberty and menopause. The periods of puberty and menopause are based on observable phenotypes. Studies based on the observed phenotype may not correctly reflect the inner physiological changes in the body. Similarly, in animal studies, the stated time of application of ovariectomy before puberty or after puberty may not accurately represent the life stage of the animals. Therefore, it is essential to understand the molecular and physiological changes during the transition of the life stages before applying procedures that change the level of estrogen. 

It is known that there is considerable variation in the ages of puberty and menopause among women [8,12] from different ethnic groups and within the same population. Because the hormonal levels, physiological conditions, and genomic components of individuals in the human population may vary greatly from one to another, if the estrogen level is utilized as one of the factors to regulate the progress of either body growth or reproductive life stages, the protocol has to be based on each person’s profile. Thus, the personalized manipulation of the estrogen levels at different life stages in humans is essential. Unfortunately, in studies using animal models, systematic investigations of the variations in puberty and menopause have not been conducted. The lack of understanding of the variations in puberty and menopause in different strains or species of laboratory animals is a major obstacle for the study of the manipulation of estrogen levels.

### 4.2. Systematically Testing the Effects of Estrogen Based on the PLOSP Hypothesis

While many questions remain, it is critical to identify suitable methods to test the key question of whether the changes in estrogen level at different life stages will have different effects on the lifespan. 

Animal studies with ovariectomy at different ages have never been systematically compared. If such a study is planned, common and normal strains should be used because the relevant pathways brought on by the ovariectomy may interact with other pathways or genetic variations. As we have discussed, the previous studies with different strains or mutations have shown different results [19,23]. In particular, for animal studies using mouse and rat models, the key is to make sure that the application of ovariectomy in occurs in different life stages [26]. At present, the exact translational periods between different life stages in animal models are not completely understood. Therefore, if one wants to perform the ovariectomy before puberty to test whether reductions in estrogen at body’s growth stage may extend the lifespan in animal models, the time of ovariectomy should be far away, e.g., several days before the traditionally known time of puberty for the animals (Figure 4). 

As the ovariectomy surgery may cause damage to the whole body and affect the growth and development, alternative approaches may be taken to investigate the effects of estrogen on the different life stages. These approaches include the use of unilateral ovariectomy, a fiber-rich diet, and chemical and drug treatments [34,35]. As our knowledge on the whole genome sequences and molecular pathways rapidly increases, approaches at genetic and molecular levels may also be possible in the near future [36]. 

In humans, before a safe protocol or medical treatment is developed, such a test as ovariectomy is improper. However, monkeys are suitable for the study of the effects of ovariectomy. Currently, these studies have focused on the impacts of ovariectomy on the physiological and pathological effects of ovariectomy. A well-designed study on the effects of the estrogen levels at different life stages will greatly benefit the treatment for personalized longevity. Similar to the animal study, as bilateral ovariectomy may impose great damage to the health conditions of the body, other methods such as unilateral ovariectomy or drugs may be used in the investigation when using mammals such as monkeys [34,35,36].

## 5. Conclusions: Integration and Personalization of Estrogen Levels for Individual Life Stages

Although the estrogen level represents the changes of life stages in women, the regulation of life stages is very complicated, involving genetic control, tremendous physiological and metabolic modifications, and significant environment influences. In this regard, changes in estrogen level do not equal changes in life stages. Thus, the manipulation of the level of estrogen must be integrated with other factors. Otherwise, the complications may lead to a decreased rather than increased lifespan [37]. To achieve an integrated approach to the regulation of estrogen levels and the extension of life stages, many more studies are needed to understand the mechanism of the changes of life stages. In particular, it is essential to understand the genomic, physiological, metabolic, and endocrinological changes during the transition periods from pre-puberty to post-puberty, from the reproductive stage to the aging stage caused by menopause [12]. 

There is great variation in the levels of estrogen among not only different geographical regions, countries, and ethnic groups, but also individuals in the same population [1,38]. To some degree, the high and low levels of estrogen are relative, and its function depends not only on its level but also the balance with other hormones and the overall genomic and physiological condition of an individual. Therefore, the regulation of the levels of estrogen at different life stages should be based on each individual’s overall health condition at the right time and right estrogen level.

## Figures and Tables

**Figure 1 jpm-12-01995-f001:**
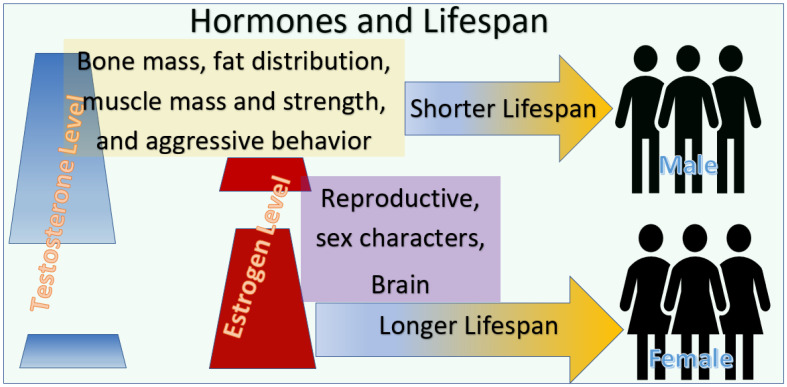
A simplified model of the levels of hormones and lifespans between sexes. Males have a shorter lifespan than females. The significant difference between the males and females is that the level of testosterone level in males is much higher than in females and the estrogen level in females is much higher than that in males.

**Figure 2 jpm-12-01995-f002:**
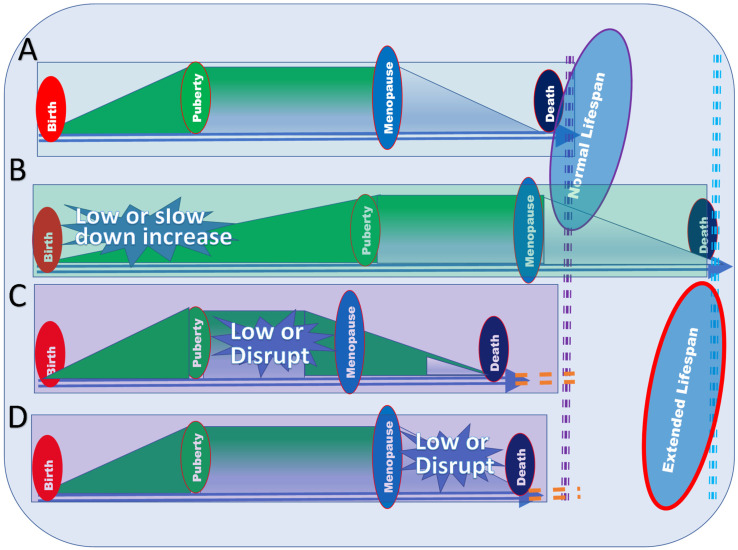
Hypothetic effects of estrogen in different life stages on females: (**A**) the life stages and normal lifespan in females; (**B**) the extension of the growth and reproductive life stages by lowering or slowing down the increase in estrogen before puberty; (**C**) the shortening of the reproductive life stage and lifespan using oophorectomy after puberty and before menopause; (**D**) the elimination of the production of estrogen and shortening of the lifespan using oophorectomy performed after menopause.

**Figure 3 jpm-12-01995-f003:**
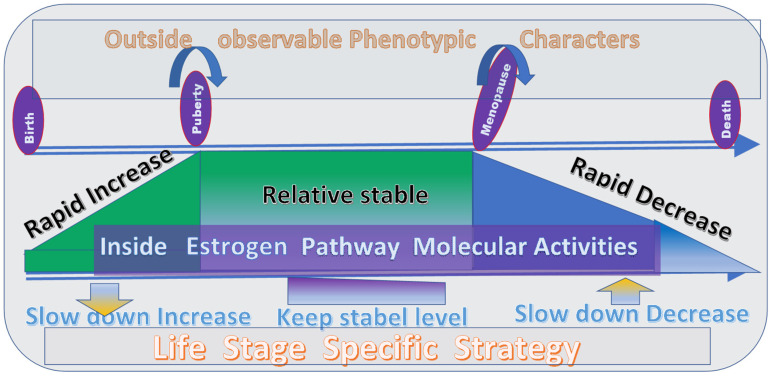
Life stages and strategies to extend the female lifespan. The upper section shows the phenotypic changes at different life stages. The middle section shows the physiological and biological alterations inside the body during the transition of different life stages. The bottom section shows the proposed life-stage-specific strategies to extend the lifespan of males.

**Figure 4 jpm-12-01995-f004:**
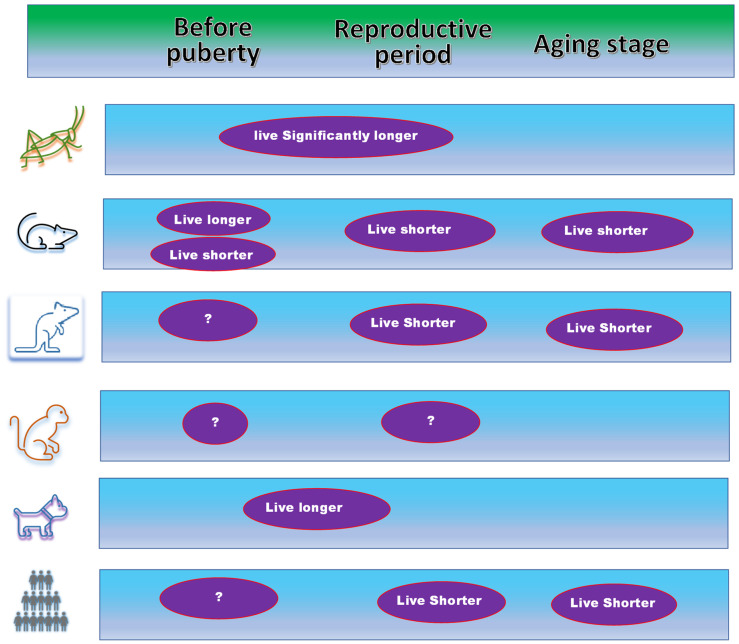
Current status of the studies of estrogen levels with oophorectomy in animal and human populations. Oophorectomy performed during reproductive and aging stages causes complications and leads to a shorter lifespan in mouse, rat, and human populations. Oophorectomy in grasshoppers extends the lifespan. In dogs, the lifespan is also extended or is not shortened when the oophorectomy is performed most likely before puberty. There are conflict reports on the impact on the lifespan when the oophorectomy is performed before puberty. More studies should be done with animals, in particular with monkeys.

**Table 1 jpm-12-01995-t001:** Examples of the effects of ovariectomy at different stages and effects in mouse and rat models.

Species	Time of Conduct	Approximate Human Age Equivalents * [17,18]	Effect on Lifespan	Reference # and Note
Mice	One month, after the first estrous cycle; 5 months of age	14 years of age and 30 years of age	Shortens the life span	[19]
	Thirteen months of age	60 years of age	Immunosenescence and oxi-inflamm-ageing	[20]
	At postnatal day 50	18 years of age	Correlation of the estrogen levels and the disease progression including lifespan	[21]
	Two weeks prior to behavioral testing (at ages of 5,17, and 22 months), mice were ovariectomized	30, 65, and 68 years of ages	Shortened lifespan in comparison with intact animals.	[22]
	At 4 weeks, bilaterally At 3 weeks, bilateral ovariectomies	13 years of age	Prolongs survival time	[23]
	-	11 years of age	Control/Ox = 265.6/230.5	[24] With CBA strain.
Rat	3 and 18 months	7 and 45 years of age	Inducing complications	[25]
	At 23 weeks of age	57 years of age	Alterations in body weight and energy metabolism induced by ovariectomy and high-fat diet consumption might not directly affect the lifespan of female rats.	[26]

* Ages equivalent to humans are estimated according to references [17,18].

## Data Availability

Not applicable.

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
