# Peer review of "Level of Estrogen in Females—The Different Impacts at Different Life Stages"

_jpm, 2022, doi:10.3390/jpm12121995_

Round 1
Reviewer 1 Report
Dear authors,
Please find below some suggestions that may improve the quality of the text.
Abstract. Please insert an objective and a brief description of the method used to achieve it.
We suggest adding a sentence on estrogen and quality of life.
Introduction. And the other sections? Is the whole text an introduction?
Typo: "human population" (2nd paragraph)
Please indicate the origin of this "principal law of lifespan (PLOSP)" idea.
Question: why would someone perform ovariectomy before puberty? Tumors? Differences in sex development (DSD)? Or are you trying to develop a theory for poeple with gonadal dysgenesis? Please make a brief statement on the rationale.
Sentence "Because this phenomenon comes from a sex difference, the reason for such a difference is automatically tracked to the feature of sex hormones" and Figure 1. This model is extremely simplified. One should take into consideration many other gender-associated risk factors, specially behaviorial aspects (smoking, BMI, physical exercise, alcohol and driving, habit of attending medical appointments...).
Page 3, line 1. "early menopause" is not a symptom, is a clinical diagnosis based on many different symptoms, including hot flushes and irregular menstruation.
Typo: "stages" (page 4, 5th paragraph).
How do you feel about replacing "castration" by "orchiectomy"? Since ovariectomy may also mean castration, one should specify which gonads are being removed.
What do you mean by saying that "Accordingly, personalized manipulation of estrogen level at different life stages in humans is necessary"?
Please insert a context to "if one wants to perform the ovariectomy before puberty"... why someone would do this?
The data you present does not allow us to say "regulation of levels of estrogen for different life stages should be individualized".
Good luck!
Author Response
Dear Reviewer, we appreciate very much for your kindly reviewing of this manuscript. We specially thank all your suggestions and comments, which are extremely helpful. Accordingly, we have addressed every comment below, and made considerable changes in the manuscript. We sincerely hope the manuscript reaches to the level for publication now.
- Abstract. Please insert an objective and a brief description of the method used to achieve it.
We suggest adding a sentence on estrogen and quality of life.
1A. Thank for your suggestion. We did not include an objective and method, or conclusion because the manuscript is a perspective, and we explore a new insight of the research on the estrogen.
As you have suggested, we have included the following sentences into the abstract:
“The objective of this article is to provide readers with a new vision of the research on the estrogen”.
“Based on the PLOSP, we recaptured the estrogen levels at different life stages and explored potential alternative approaches on the manipulation of the levels of estrogen based on the biological features of difference life stages.”
“Estrogen is known responsible for the development and regulation of the female reproductive system and secondary sex characteristics”.
“Life quality can be measured in many ways but mainly it is how an individual is doing on the healthy, comfortable, and able to participate in or enjoy life experiences”.
- Introduction. And the other sections? Is the whole text an introduction?
2A. Sorry for the mistake. We have deleted the #1 for the introduction.
- Typo: "human population" (2nd paragraph)
3A. -corrected.
- Please indicate the origin of this "principal law of lifespan (PLOSP)" idea.
4A. Yes. You are correct. We need to first briefly introduce the principal law of lifespan (PLOSP). The following sentences have been added in front of the third paragraph in the section of “Introduction”.
“A new theory called the principal law of lifespan (PLOSP) stated that every life stage has its own specific physiological and metabolic characteristics. Each life stage can be lengthened by either slowing its processes or continuously maintaining the activities of its function.”
- Question: why would someone perform ovariectomy before puberty? Tumors? Differences in sex development (DSD)? Or are you trying to develop a theory for poeple with gonadal dysgenesis? Please make a brief statement on the rationale.
5A. Very good question. Ovariectomy has been performed before puberty in the study of animal models. It may never be performed on humans before puberty. The level of estrogen, however, may be manipulated in any life stage of humans. The manipulation will not be limited to correct the hormonal disorders but to extend the body growing stage. Thank you for bringing this issue up. We have made changes in the section “Explanation of effect of estrogen at different life stages based on PLOSP” to clarify this issue.
Changes of the sentences are:
“Thus, ovariectomy as a technique to manipulate the levels of estrogen in the study of animal models must consider the life stages.”
Studies have shown that the ages of puberty of humans have been decreased among all ethnic and geographic human populations, partially because of changes in environment and nutrition [11].
“It is expected that animal studies will show that slowing down the speed of increasing of hormone level during body growth stage will extend the length of period between the birth and puberty. In other word, it will increase the length of life stage of the body growth. The data then can be utilized in human population to manipulate the levels of estrogen and other factors to extend the body growth stage”.
- Sentence "Because this phenomenon comes from a sex difference, the reason for such a difference is automatically tracked to the feature of sex hormones" and Figure 1. This model is extremely simplified. One should take into consideration many other gender-associated risk factors, specially behaviorial aspects (smoking, BMI, physical exercise, alcohol and driving, habit of attending medical appointments...).
6A. Yes. It is a simplified figure. We also agree with you that many other factors may influence the difference in lifespan between sexes.
We should have made it clear at early time, this idea arisen decades age when the research was at early stage and resources were limited.
We have modified the sentence as the following: “Because this phenomenon comes from a sex difference, decade ago, when the research was at early stage and resources and data were limited, the reason for such a difference is automatically tracked to the feature of sex hormones”.
While we know that many other factors influence the sex difference in lifespan, we also believe that the hormones play a critical role. Many behavioral differences such as aggressiveness, physical and sociological differences are related to the sex difference. We therefore modified the sentences in the section of “The observation sex difference in lifespan” to make it clear on this issue.
We also indicated in figure 1 that the model is a simplified one to emphasize the hormonal difference in humans.
- Page 3, line 1. "early menopause" is not a symptom, is a clinical diagnosis based on many different symptoms, including hot flushes and irregular menstruation.
7A. We have made changes on this sentence as below.
“The complication of ovariectomy of adult women is described as rapid aging [12], including early menopause, depression, heart diseases, memory problems and osteoporosis”.
Typo: "stages" (page 4, 5th paragraph). -corrected
- How do you feel about replacing "orchiectomy" by "orchiectomy"? Since ovariectomy may also mean orchiectomy, one should specify which gonads are being removed.
8A. Good suggestion. We have replaced the “orchiectomy" by "orchiectomy", although in this article the difference between ovariectomy and orchiectomy seems clear to the readers. The parallel between the ovariectomy and orchiectomy sound better.
- What do you mean by saying that "Accordingly, personalized manipulation of estrogen level at different life stages in humans is necessary"?
9A. We should explain better on this. We have inserted a paragraph to explain this sentence.
“Because the hormonal levels, physiological conditions and genomic components of individuals in human population may vary greatly from one to another, if the estrogen level is utilized as one of the factors to regulate the progress of either body growth or reproductive life stages, the protocol has to be made based on each person’s profile. Thus, personalized manipulation of estrogen level at different life stages in humans is necessary.”
Accordingly, personalized manipulation of estrogen level at different life stages in humans is necessary
- Please insert a context to "if one wants to perform the ovariectomy before puberty"... why someone would do this?
10A. We are sorry for the misunderstanding. We were talking about animal studies. We have modified the sentence to clarify the confusion.
“Therefore, if one wants to perform the ovariectomy before puberty to test whether re-duction estrogen at body growth stage may extend the lifespan with animal model, the time of ovariectomy should be far away”.
- The data you present does not allow us to say "regulation of levels of estrogen for different life stages should be individualized".
11A. How about to change it into this one:
“Therefore, regulation of levels of estrogen at different life stages should be based on each individual’s overall health condition at the right time and right level.”.
Thanks so much,
Weikuan
Reviewer 2 Report
Dear Author'
In the manuscript, you emphasized that "the effect of the ovary may be greater than that of the testicle on the development of the human body" but I think you should explain more about the molecular mechanisms of this claim. Why does estrogen increase lifespan? please explain in the separate section about molecular mechanisms and biological effects of estrogen in different levels of concentration and in comparison with testosterone in healthy humans and it is better to make a picture about this.
Author Response
In the manuscript, you emphasized that "the effect of the ovary may be greater than that of the testicle on the development of the human body" but I think you should explain more about the molecular mechanisms of this claim. Why does estrogen increase lifespan? please explain in the separate section about molecular mechanisms and biological effects of estrogen in different levels of concentration and in comparison with testosterone in healthy humans and it is better to make a picture about this.
A: Thank you for your kind comments to our manuscript. You made an important point. We have modified the figure 1 to explain why people believe that the estrogen makes women live longer than the men, in the opposite, why the androgen makes men live shorter than the women.
In addition, we have inserted the following sentences have been inserted into the Section of “The observation sex difference in lifespan”:
“One risk factor related to the testosterone for lifespan in male is that it is connected to the aggressiveness”
“Because this phenomenon comes from a sex difference, decade ago, when the research was at early stage and resources and data were limited, the reason for such a difference is automatically tracked to the feature of sex hormone while the molecular mechanism is still not completely understood”.
“The function of estrogen on brain is mainly cause for brain function with anxiety and depression, not the aggression.”
Thanks very much,
Weikuan
Reviewer 3 Report
Review: Level of Estrogen in Females-The Different Impacts at Different Life Stages
Overall: The study of estrogen and lifespan is important and it is commendable that the authors attempt to summarize what is known about this topic.
However, the whole manuscript has some grammatical issues and may benefit from help with appropriate grammar/word usage. This includes switching between past and present tenses inappropriately and word choices are not always appropriate.
It is also disorganized and difficult to follow, though this may be partially related to the grammar issues mentioned above.
Abstract: For the reasons listed above the abstract is confusing. The sentence “ovariectomy does not equal to the reduction of estrogen” doesn’t make sense when a few sentences later it says it does. I think the authors are having difficulty communicating what they mean.
It is difficult to follow due to a large number of grammar/word usage issues, I would say making it almost impossible to understand.
Specific concerns
1. I don’t understand the subdivision. It looks like there’s an “introduction” and everything else is part of the introduction which doesn’t make sense.
2. They bring up the principal law of lifespan a lot, but do not explain it. It may be beneficial to speak to why it is so important to readers as it is referenced so much.
3. On the section of data from animal models, table 1, it may be helpful to characterize the time of conduct in human staging equivalents (ie how many months is considered prepubertal in a mouse vs puberty vs fully pubertal).
4. It is confusing for me that there’s a section on animal models such as grasshopper under “confusing data from animal models” and then another section on grasshopper later. Is this a repeat of the same data, or is it different data and why does it belong in different sections
5. The section on the first paragraph of “Complicated effect from ovariectomy” I read multiple times and did not understand, likely again partially due to grammar/word issues but overall it was meandering and the point unclear.
6. It is unclear under “manipulation of estrogen level at different life stages based on PLOSP” whether the authors are actually suggesting performing ovariectomy on people before puberty which would obviously not be wise.
7. Of note, the paper doesn’t really talk about the normal fluctuations of estrogen during the menstrual cycle, it seems to treat estrogen as a single level, moving up or down, which is likely an oversimplification. Other hormones are mentioned but only in the conclusion. In my mind, it is
Difficult to just talk about estrogen without discussing the hormones that regulate it. It make some sense when you are speaking of ovariectomies to speak of them as a single entity because that is a procedure with specific outcomes, but when ovariectomy is not being referenced, estrogen and estrogen levels are referred to in isolation and really this isn’t physiologically accurate.
8. Under “The hormone replacement therapy in human populations PLOSP” it should be noted that studies have also shown some concerning outcomes with HRT, it is not ethical to only mention positive outcomes.
9. Under “correctly interpretation the estrogen level with PLOSP” (which on its own is poorly constructed) they discuss a lack of studies about physiological conditions and biomarkers, it may be helpful to mention which ones.
Author Response
Dear Reviewer, we appreciate very much for your kindly reviewing of this manuscript. We specially thank all your suggestions and comments, which are extremely helpful. We apologize for the confusions because of disorganization and unclarity in the manuscript. As you can see, we have made considerable changes in the manuscript including origination of the manuscript and grammar checking. We tried our best to answer your questions. We try to make sure to use the past tenses when directly cite the results of publications and to use present tenses in the rest of the manuscript. This manuscript is one of the series of publications on the theory of PLOSP. We apologize for not explain the PLOSP well at the beginning of the manuscript. We sincerely hope the manuscript reaches to the level for publication now.
I am sorry the format of questions and answers was changed and please be patient to read through.
Overall: The study of estrogen and lifespan is important and it is commendable that the authors attempt to summarize what is known about this topic.
- However, the whole manuscript has some grammatical issues and may benefit from help with appropriate grammar/word usage. This includes switching between past and present tenses inappropriately and word choices are not always appropriate.
It is also disorganized and difficult to follow, though this may be partially related to the grammar issues mentioned above.
- Your point is important to us. We have gone through the manuscript and made considerable changes including the organization of the manuscript and grammar and word usages.
- Abstract: For the reasons listed above the abstract is confusing. The sentence “ovariectomy does not equal to the reduction of estrogen” doesn’t make sense when a few sentences later it says it does. I think the authors are having difficulty communicating what they mean.
It is difficult to follow due to a large number of grammar/word usage issues, I would say making it almost impossible to understand.
- We are sorry about the confusion caused to you. We intend to say that the effect of ovariectomy is much more than the only reduction of level of estrogen. As this aspect is not the focus of the manuscript and it causes the confusion. We have eliminated this partial sentence.
Specific concerns
- I don’t understand the subdivision. It looks like there’s an “introduction” and everything else is part of the introduction which doesn’t make sense.
- Sorry for the mistake. We have deleted the #1 for the introduction. The manuscript has been reorganized.
- They bring up the principal law of lifespan a lot, but do not explain it. It may be beneficial to speak to why it is so important to readers as it is referenced so much.
- Yes. You are correct. We need to first briefly introduce the principal law of lifespan (PLOSP). We should not assume that the reviewer would have already known this new theory. The following sentences have been added in front of the third paragraph in the section of “Introduction”.
“A new theory called the principal law of lifespan (PLOSP) stated that every life stage has its own specific physiological and metabolic characteristics. Each life stage can be lengthened by either slowing its processes or continuously maintaining the activities of its function.”
- On the section of data from animal models, table 1, it may be helpful to characterize the time of conduct in human staging equivalents (ie how many months is considered prepubertal in a mouse vs puberty vs fully pubertal).
- Thanks for the nice suggestion. We have added the approximate human ages in a column. We also added two references for these estimations. The references have been reorganized accordingly.
- It is confusing for me that there’s a section on animal models such as grasshopper under “confusing data from animal models” and then another section on grasshopper later. Is this a repeat of the same data, or is it different data and why does it belong in different sections
- It is true this section caused confusion. The original intention is to emphasize the difference between grasshopper and mice/rat model. We have removed this section and connected the two sentences into the prior section.
- The section on the first paragraph of “Complicated effect from ovariectomy” I read multiple times and did not understand, likely again partially due to grammar/word issues but overall it was meandering and the point unclear.
- The intension of this section is to discuss the effect of ovariectomy and its complicated effect on the level of estrogen, other hormones and damage to the body by the surgery itself. However, the title is not clear and lack of brief introduction of the effect of ovariectomy. We have modified the title of this section and added few sentences before the first paragraph to introduce the complicated effect and then discuss the causes of these complicated effect.
The tile for this section has been changed to “Difference between ovariectomy and decrease of estrogen level”
An introduction below has been put before the first paragraph.
“As shown above, the studies of ovariectomy in animal models have produced controversial results. Its effect on the lifespan is not clear. In human population, ovariectomy are known causing multiple side effect such as menopause signs and symptoms, depression or anxiety, osteoporosis etc. Overall, the effect of the ovariectomy seems complicated”. However, the effect of ovariectomy is much more than a depletion of estrogen production”.
- It is unclear under “manipulation of estrogen level at different life stages based on PLOSP” whether the authors are actually suggesting performing ovariectomy on people before puberty which would obviously not be wise.
- You are right. No. We are not suggesting performing ovariectomy before puberty on humans. However, ovariectomy can be used as one of the tools to test the effect of estrogen in animal model. Based on the data from animal study, any study on humans on the manipulation of level of estrogen has to be done in other ways, mostly perhaps in the form of drugs or nutrition supplements.
Your question reminder us that we have to make sure that readers will not misunderstand what we present in this section. We have done through this section and made changes whenever possible to clarify this issue. Sentences have been inserted whenever necessary. Examples are below:
“While the ovariectomy can be used for the studies with animal models to test the effect of estrogen and other hormones, for humans, other means must be taken for such a study. Manipulation of estrogen level using molecular approach is most likely a suitable approach”.
- Of note, the paper doesn’t really talk about the normal fluctuations of estrogen during the menstrual cycle, it seems to treat estrogen as a single level, moving up or down, which is likely an oversimplification. Other hormones are mentioned but only in the conclusion. In my mind, it is Difficult to just talk about estrogen without discussing the hormones that regulate it. It make some sense when you are speaking of ovariectomies to speak of them as a single entity because that is a procedure with specific outcomes, but when ovariectomy is not being referenced, estrogen and estrogen levels are referred to in isolation and really this isn’t physiologically accurate.
- Thanks for your reminding us about to clarify this issue. When we talk about the estrogen, we are talking about the group of hormones that play an important role in the normal sexual and reproductive development in women. As you noticed, we did not intend to talk about the normal fluctuations of estrogen during the menstrual cycle, but we treat the estrogen as a term of group of hormones that regulates sexual and reproductive development in women.
In consideration of your concern, we have inserted a sentence in the first paragraph of the introduction to make it clear.
“Estrogens are a group of hormones that play an important role in the normal sexual and reproductive development in women”.
In addition, under section of “Confused data from animal models”, we inserted one sentence to explain that ovariectomy has been used in animal models as a tool to test the effect of estrogen.
“Thus, ovariectomy has been used in animal models as a tool to test the effect of the group of hormones of estrogen.”
- Under “The hormone replacement therapy in human populations PLOSP” it should be noted that studies have also shown some concerning outcomes with HRT, it is not ethical to only mention positive outcomes.
- Great point. We have added a paragraph to talk about the side effects of the HRT and the proper manipulation of the hormone level.
“Currently, HRT also causes multiple side effects. While in general every therapeutic application has side effects, it is possible to reduce the side effects of HRT with improved protocols in the future, such as the individualized application of HRT with different protocols. For example, recently it has been confirmed that thyroid hormone replacement therapy was effective in patients with heart failure and low-triiodothyronine syndrome [33]. Despite of the side effect, that fact that overall, the increased lifespan by HRT shows that regulation of hormone levels in the right time and right life stage may be sued as one tool to extend the lifespan”.
- Under “correctly interpretation the estrogen level with PLOSP” (which on its own is poorly constructed) they discuss a lack of studies about physiological conditions and biomarkers, it may be helpful to mention which ones.
- The title of this section has been changed to “Correctly interpretation the estrogen level based on PLOSP”.
This statement is focused on lack of studies for “life stages transition and turning point between the life stages”. We did not find any publication on the comparison of the transitions between the life stages. In consideration of your comments, we have inserted the following sentence to further explain our point.
“At present, we did not find any study to compare the life stage transition and turning point between the life stages, such as the comparison of physiological alterations during the transition periods between the puberty and the menopause”.
“Because the hormonal levels, physiological conditions and genomic components of individuals in human population may vary greatly from one to another, if the estrogen level is utilized as one of the factors to regulate the progress of either body growth or reproductive life stages, the protocol has to be made based on each person’s profile. Thus, personalized manipulation of estrogen level at different life stages in humans is essential.”
Round 2
Reviewer 1 Report
Authors provided the changes required.